# Interactions between Bacteriophage, Bacteria, and the Mammalian Immune System

**DOI:** 10.3390/v11010010

**Published:** 2018-12-25

**Authors:** Jonas D. Van Belleghem, Krystyna Dąbrowska, Mario Vaneechoutte, Jeremy J. Barr, Paul L. Bollyky

**Affiliations:** 1Laboratory Bacteriology Research, Department of Clinical Chemistry, Microbiology and Immunology, Ghent University, 9000 Ghent, Belgium; Mario.vaneechoutte@ugent.be; 2Division of Infectious Diseases and Geographic Medicine, Department of Medicine, Stanford University School of Medicine, Stanford, CA 94305, USA; pbollyky@stanford.edu; 3Bacteriophage Laboratory, Institute of Immunology and Experimental Therapy, Polish Academy of Sciences, 53-114 Wrocław, Poland; dabrok@iitd.pan.wroc.pl; 4School of Biological Sciences, Monash University, Melbourne, VIC 3800, Australia; jeremybarr85@gmail.com

**Keywords:** bacteriophage, immunology, innate immunity, adaptive immunity, human host, phage-human host interaction

## Abstract

The human body is host to large numbers of bacteriophages (phages)–a diverse group of bacterial viruses that infect bacteria. Phage were previously regarded as bystanders that only impacted immunity indirectly via effects on the mammalian microbiome. However, it has become clear that phages also impact immunity directly, in ways that are typically anti-inflammatory. Phages can modulate innate immunity via phagocytosis and cytokine responses, but also impact adaptive immunity via effects on antibody production and effector polarization. Phages may thereby have profound effects on the outcome of bacterial infections by modulating the immune response. In this review we highlight the diverse ways in which phages interact with human cells. We present a computational model for predicting these complex and dynamic interactions. These models predict that the phageome may play important roles in shaping mammalian-bacterial interactions.

## 1. Introduction

Commensal microorganisms colonize and live in symbiosis with the human body and encompass diverse phyla from the three domains of life: Eukarya, Archaea, and Bacteria. Body surfaces that are in direct contact with the environment, including the intestine, skin, urogenital tract, and upper respiratory tract harbor most of these microorganisms. The bacterial component of the human microbiota and its associated genes have been a primary focus of research efforts over the past two decades [1,2,3]. These efforts have yielded a wealth of insight about the composition of human-associated bacterial communities, how these resident bacteria interact with the immune system and how bacterial-immune system interactions are altered in disease [1,4,5].

The microbiota of healthy humans also includes a large number of bacterial viruses, or bacteriophages (phages) [6]. Phages were previously regarded as bystanders that only impacted immunity indirectly via effects on the mammalian microbiome. However, it is becoming clearer that phages also impact immunity directly. 

In this review we highlight the diverse ways in which phages interact with human cells: [1] effect of phages in the mammalian interface, [2] innate immune response, and [3] the adaptive immune response against the phages. We then present a computational model for predicting these complex and dynamic interactions. This model predicts that our phageome may play important roles in shaping mammalian-bacterial interactions, underlying the important effect of phage induced anti-inflammatory properties. Finally, the gaps in our knowledge and potential future lines of investigation are highlighted.

## 2. The Human Phageome

Phages colonize all body niches, including the skin [7,8], oral cavity [9,10,11], lungs [12,13,14], gut [15,16], and urinary tract [17]. However, phages are frequently overlooked in microbiome and metagenomic studies and their role is often unclear. Most phages present in these viromes are temperate phages that can integrate their DNA into the bacterial genomes (i.e., prophage) or be present as episomes, and as such can alter the phenotype of the host bacteria by lysogenic conversion [16,18]. Although human blood is considered to be sterile, metagenomic analysis has shown the presence of a viral community, most of which belonged to the *Myoviridae, Podoviridae, Siphoviridae, Microviridae,* and *Inoviridae* families [19,20,21,22]. Once present in the blood, these phages may interact with immune cells and induce innate and adaptive immune responses [23,24,25,26].

Of all the microbial communities within the body, the intestinal community is by far the most complex and dense. The human gut microbiome, as shown by metagenomic studies, includes many viral genes (the virome) [15,16,27,28]. Approximately 90% of the gut virome consists of phages [29], estimated at 10^9^ viruses per gram of feces [30,31]. As new members of the bacterial community are introduced, the phage populations in the intestine diversify, suggesting that phage diversity and bacterial diversity are linked [32]. Furthermore, this relationship is very dynamic in infants and stabilizes in adults [33]. Although there is less variation of intestinal phage populations within individuals over time, there is substantial variation between individuals, even when those individuals have similar bacterial community structures [15,16].

Phages can supply bacteria with genes that are involved in toxin, polysaccharide, and carbohydrate metabolism, and, in rare cases, they represent a source of antibiotic resistance [34,35]. Some phages can modulate bacterial antigenicity through the production of enzymes capable of modifying the O-antigen component of LPS in microorganisms such as *Escherichia* coli, Salmonella spp., *Shigella* spp., and *Vibrio cholerae* [36,37,38,39].

It is thus important to consider whether phage interactions with commensal bacteria could alter community compositions in ways that impact the function of the immune system and influence the spread of pathogenic viruses, or even bacteria [1,40,41,42]. Among the mechanisms responsible for the recognition of microbial and viral structures are the Toll-like receptors (TLR) [43]. These TLR are able to recognize Pathogen Associated Molecular Patterns (PAMPs) (e.g., LPS, flagellin, or unmethylated CpG-DNA). Viral nucleic acids can be recognized by multiple TLR, notably TLR9 recognizes DNA, whereas TLR7 and 8 recognize ssRNA and TLR3 recognizes dsRNA [44,45,46]. These nucleic acid-sensing TLRs have the potential to promote, amongst others, the production of Type I IFN.

The virome continuously stimulates low-level immune responses without causing any overt symptoms [47,48]. Duerkop and Hooper hypothesized that commensal bacteriophages could activate one or more innate immune pathways, thereby stimulating antiviral immune responses and continuously inducing low cytokine production. These cytokines also exert their action on non-immune cells and may continuously induce inflammatory processes, thereby conferring constant protection against pathogenic viral infections [1,49].

It is clear that phages are omnipresent and form a major constituent of many microbiomes, nevertheless the interactions of phages with their human host warrants further research.

## 3. Phages Effects on the Bacterial - Mammalian Host Interface

### 3.1. Phages and Mucosal Tissues

Phages interact with host immunity at the mucosal surface. The mucosal surface (e.g., the human gut and respiratory tract) represents a critical immunological and physiological barrier within all animals that both protects against invading bacterial pathogens while also supporting large communities of commensal microorganisms [50,51]. The mucosal surface is predominantly composed of mucin glycoproteins that are secreted by the underlying epithelium. By offering both structure and nutrients, mucus layers influence the composition of the microbiota and select for commensal symbionts [52,53,54]. It has been shown that mucosal surfaces of the gut commonly support more abundant and stable bacterial populations than the surrounding environments (e.g., the luminal content of the gut) [55,56]. This is, in part, due to the degradation of mucins by gut microbes, but also in part due to host epithelial secretions that selectively shape the commensal microbiota [53,54,57]. These host secretions are diverse and can include antimicrobials, such as alpha-defensin and RegIIIϒ [58,59]. Conversely, when mucosal surfaces are invaded by pathogenic bacterial species, the epithelium may respond by increasing the production of antimicrobial agents, hypersecretion of mucin, or alteration of mucin glycosylation patterns in an attempt to subvert microbial attachment and to increase physical removal of the invading bacterial species [60,61,62].

These mucosal layers also harbor large and diverse communities of phages (Figure 1A). Mucus-associated phage communities are significantly enriched compared to the surrounding non-mucosal environment [63]. Investigations across diverse mucosal surfaces ranging from those present in corals, fish, mice, and humans revealed an average 4.4-fold increase in phage numbers in mucus relative to bacterial cells [63,64,65]. This increase in phage abundance happens through an adherence mechanism whereby phages weakly bind mucin glycoproteins via immunoglobulin-like (Ig-like) protein domains displayed on their capsids. The Ig-like fold is one of the most common and widely dispersed in nature, present in antibodies and T-cell receptors where it mediates important binding interactions of the human adaptive immune system [66,67]. These Ig-like domains are found within approximately one quarter of sequenced *Caudovirales* genomes, and are typically displayed on the virion surface [68,69]. Most of these structurally displayed Ig-like domains are dispensable for phage growth in the laboratory, which led to the hypothesis that they aid the phage in the adsorption to their bacterial host under environmental conditions [68,70]. Phages that utilized Ig-like domains, which effectively bind to the mucus layer, would be under positive selection within the mucosa, leading to the proposal of a bacteriophage adherence to mucus (BAM) model as a non-host-derived layer of immunity, mediated by phages [63,71].

On top of their direct effects on bacterial populations, phages can also have an indirect effect on the colonization of their bacterial host to mammalian cells. In case of *Neisseria meningitidis* it has been shown that its filamentous phage (MDAφ) increases its host-cell colonization [73]. The authors showed that the presence of this filamentous phage leads to a higher binding of the bacteria to the host epithelial cells. Furthermore, the phage also seemed to form a linker between the bacteria, further heightening its colonization. These effects were not observed for endothelial cells, indicating a specificity of the phage towards epithelial cells. In this case it is the phage itself that forms an additional virulence factor to the bacteria, promoting bacterial aggregation.

It can be further hypothesized whether there is a mutual benefit to phage and bacteria, whereby the phage interacts with the mucosal surface and binds the bacteria. Instead of infecting and lysing the bacteria, the phage would provide the bacteria with additional binding sites, thus, elevating the colonization frequency.

### 3.2. Phage Transcytosis

Below the mucosal surfaces, the cellular epithelium forms another physical barrier that separates the heavily colonized mucosa from the normally sterile regions of the body. Due to their ubiquity within the epithelial mucus layer, phages are in constant contact with the epithelial layers. The passage of commensal bacteria colonizing the intestine across the mucosal epithelium to local lymph nodes and internal organs is termed bacterial translocation and is a critical step in the pathology of various disorders [74,75]. While bacterial translocation is a well-described phenomenon, little is known about the translocation of bacterial viruses.

Low internalization of bacteriophages by enterocytes and other endothelial cells was demonstrated for M13 phages (empty vectors used as a control in phage display) in vivo [76] and in vitro [77]. In vitro uptake of phage M13 could be blocked by chloroquine, an inhibitor known to block clathrin-dependent endocytosis, suggesting this was the proposed pathway for internalization [77]. Since this type of endocytosis is strictly receptor-mediated (i.e., external objects must be bound to a membrane receptor to be dragged into the pits), there is reason to think that phage uptake can be a consequence of specific phage-to-epithelium interactions.

In vivo studies of oral administration of non-engineered phages demonstrated both effective [78,79,80,81,82] and ineffective [83,84,85,86,87,88] systemic dissemination. This demonstrates that natural phage translocation from gut to circulation is possible but suggests a range of other factors may regulate this process, such as physiological status of a host [24,89] and characteristics of the phage. To some extent, physical parameters of phage particles, like their size and shape, may influence the phage’s ability to penetrate mammalian bodies. However, the most important factor seems to be the dose, which correlates strongly with the probability that an orally applied phage can be found in circulation or in tissues. This is in line with the fact that phages may differ in their ability to propagate on gut bacteria and this ability may further limit their systemic dissemination after application *per os* [86,90].

An important consideration regarding the translocation of orally administered phages is whether phages can cross the mucosal barrier in sufficient numbers to subsequently interact with and bypass the cellular epithelium. Recently, it has been demonstrated that phages can enter and cross epithelial cell layers by a non-specific transcytosis mechanism [91]. Phage-epithelial transcytosis seems to preferentially occur in an apical-to-basal direction and was shown to occur across different types of epithelial cell layers (e.g., gut, lung, liver, kidney, and brain cells) and for diverse phage types and morphologies (e.g., *Myoviridae*, *Siphoviridae*, and *Podoviridae*; Figure 1B). Microscopy revealed that roughly 10% of epithelial cells endocytosed phage particles, which appeared to be localized within membrane-bound vesicles. Interestingly, those few cells that did endocytose phage particles appeared to contain large numbers of such vesicles. Chemical inhibitor assays suggest that, once endocytosed, phage particles traffic via the Golgi apparatus before being functionally exocytosed at the basal cell layer. The transcytosis of phages across epithelial cell layers provides a mechanistic explanation for the systemic occurrence of phages within the human body in the absence of disease [91]. Contrary to these observations, others have observed the accumulation of phagocytosed phages near the cell nucleus of MAC-T cells [92]. The presence of phages close to or in the nucleus reassess the question as to whether phages might be able to have their genome replicated or translated. Furthermore, these data raise the question of whether the production of phage derived RNA induces cellular responses or whether the presence of the phage close to or in the nucleus have an effect on the cellular function of the phage “infected” mammalian cell.

## 4. Cell Perfusion and Access, Interaction with Intracellular Immune Response

The penetration of phages in higher organisms leads to direct contact of phages with eukaryotic cells. Therefore, it is important to know whether these phages can interact with or infect eukaryotic cells. Infection seems unlikely, because elements of the phage tail structure only bind to specific receptors on the surfaces of their target bacteria. Furthermore, it is generally recognized that phages cannot infect eukaryotic cells, because of major differences between eukaryotes and prokaryotes in regard to key intracellular machinery that are essential for translation and replication [93]. This was illustrated by Di Giovine et al. [94], who re-engineered the filamentous phage M13 to infect mammalian cells. Although subsequent binding and internalization of the engineered phage was observed, no multiplication of the phage was detected [94]. Further engineering of filamentous phages has shown the potential of these phages to produced RNAs in eukaryotic cells after their uptake [95,96]. Although most of these systems made use of eukaryotic gene promoters to drive transcription, these data demonstrate the potential for phage derived nucleic acids to be recognized by eukaryotic cellular pathways, including TLR and other induced (viral) immune responses.

Infection aside, it is feasible that phages can directly interact with eukaryotic cells, either extra- or intra-cellularly. Nguyen et al. [91] performed cellular fractionation of epithelial cells that had been incubated with phages and showed complete perfusion of the eukaryotic cell, with phage particles seen within all endomembrane compartments. From here, phage particles are likely degraded, shuffled, and transported throughout the cell, providing ample opportunities to interact with eukaryotic cellular components. The specific mechanisms involved remain largely uninvestigated, but could conceivably include recognition or binding with phage structural proteins or recognition, binding, transcription, or translation of phage nucleic acids [97].

It has recently been demonstrated that *E. coli* phage PK1A2 can actively bind and penetrate eukaryotic neuroblastoma cells in vitro. The interaction of the phage is attributed through the binding of cell surface polysialic acid by the phage, which shares structural similarity with the bacterial phage receptor [98]. The authors were able to show that these phage particles were able to be present in these cells for up to 24 h without affecting cell viability. Uptake of these phage particles may also lead to the activation of intracellular immunity, potentially priming the eukaryotic cell into an antimicrobial state or enhancing barrier function [99]. Further research is needed within this area to elucidate intracellular phage-eukaryote interactions.

## 5. Phage Innate Immune Response

### 5.1. Phage Phagocytosis

It is well established that phages can be phagocytosed by mammalian cells [100,101,102]. As such, the immune system plays a key role in phage clearance from animal and human bodies. Elements of the mononuclear phagocyte system (MPS) in the spleen and liver filter foreign objects, including phages, from the circulation. The spleen and liver have been identified as the major sites of phage accumulation, as phage titers are usually the highest there [103]. The MPS has been credited for the rapid removal of administered wild-type phage λ from the circulatory system in humans [104]. Moreover Merril et al. [105]) were able to identify certain phage λ mutants that were capable of circumventing the MPS immune response, whereby these mutants prevailed more than 24 h longer in the blood stream of mice than the wild-type phage. These phage λ mutants contained a single Glu-Lys substitution in the phage capsid protein E, leading to a charged change [105].

Both organs contain a large fraction of professional phagocytes. Phagocytosis by immune cells within the liver and spleen seem to be the major process of bacteriophage neutralization within the human body [26,78,80,104,106,107,108]. One should note that phagocytosis allows the removal of phage particles, even when no specific response to bacteriophages has been developed. Consequently, phagocytes are probably the major fraction of animal or human cells that interact with bacteriophages in vivo.

Clear evidence concerning the cooperation of phages with the innate immune system was first provided by Tiwari et al. [109], who showed the necessity of a neutrophil-phage cooperation in the resolution of *P. aeruginosa* infections [109]. The authors demonstrated that the presence of neutrophils is necessary to remove phage resistant bacteria, which emerge during the phage therapeutic treatment when only a single phage is used. This was later repeated by Roach et al. [110] and Pincus et al. [111] and converted into an in-silico model by Leung & Weitz [112].

Studies, in vitro [23,113,114] as well as in vivo [25,115], regarding the cellular immune response induced by phages have been conducted in recent years and revealed the potential of phages to interact with the mammalian immune system (Figure 2). However, it should be noted that many experiments [113,115] concerning the immune response induced by phages have been carried out using phage lysates containing remnants of lysed bacteria (e.g., LPS, cytosolic proteins, or membrane particles) or fragments of the host bacterial cell wall adhered to phage tails. This makes it extremely difficult to determine which components were truly responsible for the modulation of the immune response.

### 5.2. Phage Induced Phagocytosis of Bacteria

Phages can also increase phagocytosis of bacteria by macrophages, since phages administered together with the host bacteria were able to stimulate bacterial phagocytosis [116] (Figure 2). This was attributed to opsonization of bacterial cells by phages, where the phage coats the bacteria and makes it more recognizable for the immune system. This opsonization is in addition to the direct lytic activity of phages, which may contribute to the effective elimination of pathogenic bacteria in vivo. As phages continue the process of infection when adsorbed onto their bacterial host, some authors have suggested that during phagocytosis, phages continue lysing the phagocytosed bacteria, helping the activity of phagocytic cells [117,118].

One of the possible responses of phagocytes to foreign objects is the production of reactive oxygen species (ROS). ROS mediate antibacterial activity of phagocytic cells, but excessive ROS production may cause oxidative stress and tissue damage. A preliminary study performed by Przerwa et al. [119] suggested that phage T4 influenced the phagocyte system and inhibited the ROS production in response to pathogenic bacteria (i.e., *Escherichia coli*). This phenomenon appeared to depend on specific phage-bacterium interactions, but the precise mechanism is currently not known. Furthermore, the host-specific effect could indicate that the ROS reduction is caused by a reduction of bacteria due to infection and lysis by the phage and not due to direct effects by the phage, per se.

A more comprehensive follow-up study was conducted, whereby polymorphonuclear leukocytes (PMN) were stimulated with one of three different R-type *E. coli* strains (i.e., *E. coli* B or *E. coli* J5, both susceptible for T4, or *E. coli* R4, resistant to T4) or with LPS derived from these three strains [120]. Through this setup, the authors could observe a reduction in ROS production when PMNs were stimulated, with either the live bacteria or their LPS in the presence of phage T4. The results provided by these authors indicate the potential of phages to directly modify functions of mammalian cells and to exert anti-inflammatory properties [120]. A possible explanation for a mechanism underlying phage ability to reduce bacteria-induced ROS production in phagocytes was proposed by Miernikiewicz et al. [120], who investigated T4 phage tail adhesin gp12, which specifically binds bacterial LPS and decreased the potency of LPS to induce an inflammatory response in vivo [121].

### 5.3. Cytokine Response against Phages

Several studies have been conducted to determine the potential of phages to induce a cytokine response. Often these studies make use of phage preparations that where not fully purified from bacterial endotoxins or proteins. For example, Park et al. [115] studied the cytokine production in mice induced by phage T7, after they were fed with a single dose of phage T7 every 24 hours for 10 days (an exact dose was not provided by the authors). The authors were able to demonstrate that phage T7 induced a very minor increase of inflammatory cytokine production in mice, although no histological changes were observed in the tissues or organs.

On the other hand, analysis of the cytokine production of mice treated intraperitoneally for 5.5 h with highly purified preparations of either whole phage T4 particles, or four phage T4 capsid proteins (i.e., gp23*, gp24*, Hoc, and Soc) showed no inflammatory mediating cytokines in mice [25].

The effect of phages on the production of TNF-α and IL-6 in human serum has also been studied, as well as the in vitro ability of blood cells to produce these cytokines in response to phage. Weber-Dąbrowska et al. [113] used blood derived from 51 patients with long-term suppurative infections of various tissues and organs caused by drug-resistant strains of bacteria. These patients were treated with phages and blood samples were collected and tested for the presence of TNF-α and IL-6. The authors were able to observe a reduction in the production of these cytokines after long-term treatment (i.e., 21 days). However, the observed normalization was likely influenced by the decreased number of pathogenic bacteria in the body following therapeutic application of the phage.

In vitro studies have indicated that phages could have anti-inflammatory properties. Using five highly purified phages targeting two different pathogens, *P. aeruginosa* and *S. aureus,* it was shown that these five phages induced comparable immune responses in PBMCs derived from healthy human donors. Anti-inflammatory markers such as suppressor of cytokine signaling 3 (SOSC3), IL-1 receptor antagonist (IL1RN), and IL-6 were similarly upregulated following treatment with the different phages [23]. The anti-inflammatory action of phages is also in line with some previous observations suggesting an immunosuppressive effect of phages in murine in vivo models of xenografts [122,123]. The anti-inflammatory characteristic of phages was further strengthened by the recent observation that another *S. aureus* phage, vB_SauM_JS25, is able to suppress LPS-induced inflammation [114]. Furthermore, the authors observed that this phage suppressed the phosphorylation of NF-κB p65. Whether this effect is due to a direct interaction of the phage with NF-κB is currently not clear. Nevertheless, these studies clearly show the potential of phages to induce anti-inflammatory properties unrelated to their antibacterial activities.

It should, however, be emphasized that the potential anti-inflammatory or immunosuppressive action of bacteriophages should not be considered as comparable to physiological effects exerted by well-known anti-inflammatory or immunosuppressive drugs. The precise mechanism as to how phages are able to induce (anti-) inflammatory responses is currently not known, although the antimicrobial effect appears to be one of the factors.

### 5.4. Phage Adaptive Immune Response

#### Anti-Phage Antibody Production

Since phages consist of tightly packed DNA or RNA and a protein coat, formed by relatively large number of proteins or repeating protein units, it appears obvious that neutralizing antibodies should be produced in individuals subjected to phage therapy or exposed to naturally occurring phages [117,124,125,126] (Figure 2). Phage immunogenicity has been employed in medicine to test for immune competence of immunodeficient patients (e.g., HIV patients) [127]. In fact, immunization (intravenous administration) with bacteriophage ϕX174 is easy and has been used extensively to diagnose and monitor primary and secondary immunodeficiencies since the 1970s, without reported adverse events, even in patients in whom prolonged circulation of the phage in the bloodstream was observed. This suggests an intrinsically low toxicity of phage ϕX174, even in patients with a compromised immune system [128,129,130].

Naturally occurring bacteriophages also induce humoral immunity. Phage-neutralizing antibodies against naturally occurring phages (i.e., not therapeutically administered) were detected in the sera of different species (e.g., mice, horse, or human) [126,131,132,133]. Evaluating the anti-phage antibody production against phage T4 in 50 healthy volunteers who had never been subjected to phage therapy nor involved in phage work showed the presence of naturally occurring phage-antibodies [126]. Of the investigated sera, 81% significantly decreased phage activity, suggesting the presence of anti-phage antibodies. In these positive sera, natural IgG antibodies specific to the phage proteins gp23*, gp24*, Hoc, and Soc were identified (Figure 3). These results demonstrate that anti-T4 phage antibodies are frequent in the human population.

Most studies suggest that it is very easy to generate phage antisera by immunization of humans or animals with phages [124,126,128,129]. Contrary to this, a safety study by Bruttin and Brüssow in 2005 administering T4 phages orally at very low doses to human patients revealed no antibody induction in phage-treated patients, potentially due to the very low doses of bacteriophages administered due to safety concerns or the lack of adjuvant. Recently, a study concerning the production of IgG, IgA, and IgM in human patients undergoing phage therapy was carried out by Żaczek et al. [134], who treated 20 patients, for an undisclosed time, with the MS-1 phage cocktail (containing three lytic *S. aureus* phages), either orally or locally [134]. For most patients, no antibodies could be detected. For the few patients that produced elevated levels of IgG or IgM, the presence of anti-phage antibodies did not translate into an unsatisfactory clinical result of the phage therapy. The low antibody production against the phage cocktail could be due to the small time-scale during which the patients were treated. On the other hand, the elevated antibody production in a few patients could be due to a previous encounter of one of the phages used in the cocktail and the presence of an immunological memory.

These reports demonstrate that the humoral response does not follow a simple scheme of induction [24,89,117,126,135]. This was further studied by Majewska et al. [24], who quantified the antibody production against a single phage (i.e., *E. coli* phage T4) in mice over a time period of 240 days [24]. Phage T4 was given orally to mice for 100 days, followed by 112 days without phage treatment. The treatment was then repeated with the same phage up to day 240. It was demonstrated that the long-term oral treatment of mice with phage T4 led to a humoral response. The authors observed that this response emerged from the secretion of IgA in the gut lumen and an IgG production in the blood. The intensity of this response and the time necessary for its induction depended on the exposure to phage antigens, which is related to the phage dose. The factor limiting phage activity in the gut was the production of specific IgA. If the secretory levels of IgA were low, phages remained present in the feces. When the IgA level increased (around day 80), there were no active phages present in the feces. On the other hand, when secretory IgA decreased with time (on day 213 it dropped to its initial levels), phages could be detected again, until phage-specific IgA levels increased again.

According to the same authors, the induction of serum IgG suggests that phages can translocate from the gut lumen to the circulation. This observation is further strengthened by recent data of transcytosing phages [91]. Furthermore, it was possible to isolate phages from murine blood after oral application of high phage doses (4 × 10^9^ pfu/ml of drinking water), and this fact correlated with phage ability to induce a long-lasting secondary immune response. Lowering the phage dose ten-fold did not induce a significant increase of the adaptive immune response, nor did it allow for detection of active phages in the circulation. Besides considering the complete phage particle as a whole, it is of interest to evaluate the immune responses induced against individual phage proteins. It was demonstrated that phage T4 Hoc protein and gp12 strongly stimulated the IgG and IgA antibody production in the blood and gut respectively, while gp23*, gp24*, and Soc induced low responses [24].

### 5.5. In-Silico Modeling of the Immune Response Towards Phages

*In-silico* models predicting phage therapeutic interventions have been developed to better understand the immune response against phages and its impact on the outcome of phage therapeutic interventions [26,112,136,137,138,139]. These models are complicated by the fact that phages are protein-based biological agents that interact with the body’s immune system, actively replicate, and even evolve during manufacture or use [140]. As such, phage applications have a vastly different pharmacology compared to conventional drugs [137,138,139,141,142]. In these mathematical models, the rate at which a bacterial population declines due to phage infection, the rate at which the phage population increases, and the levels at which they are maintained depends primarily on five parameters: the infectivity of the phage, the latency period, the burst size, the rate at which the phages are degraded or removed from the site of infection, and the bacterial growth rate. Besides these five parameters, two other variables need to be taken into account: the density of susceptible bacteria and the density of the phage [136]. In summary, these models describe phage pharmacokinetics as being analogous to the population dynamics of the phage-bacterial interaction [143], not taking into account potential interaction between bacteria and phages with the innate or adaptive immunity.

These mathematical models can be further extended to include the mammalian host response towards the phage [26]. Based on experimental data, a general scheme can be developed for the tripartite interactions between phage, bacteria, and mammalian immunity. This scheme summarizes the main reciprocal dependencies, specifically the limiting or inducing effects (Figure 4). There are three initial key assumptions on which this scheme is based. First, the innate immunity is activated by the bacteria and acts against the bacteria, but at the same time it also acts against the phage. The second assumption is that phages are not able to boost an innate immune response [25,115]. The third assumption is that the adaptive immunity specific to phages and the adaptive immunity specific to bacteria have no interfering cross-talk. This led to the development of a model with a set of immunity-representing variables; innate immunity (I), adaptive immunity specific to phages (A), and adaptive immunity specific to bacteria (B). A similar in-silico model described the outcome of a phage therapeutic intervention, taking into account the occurrence of phage resistant bacteria and a phage decay rate, which represents both the innate and adaptive immunity towards the phage [110,112]. This model indicated that neutrophils are necessary to completely clear a bacterial infection when phage resistance occurs, although it could be argued that phage resistance could partially be prevented when using a phage cocktail [109,110].

When no interaction occurs between the innate immune response (I) and the phage (P), the original Hodyra-Stefaniak model predicts a successful intervention of phages in the removal of a bacterial infection (Figure 5A) [26,112]. The inclusion of the variable for the innate immunity (I) demonstrates that the expected outcome of phage therapy could be abrogated by the innate immunity boosted by the bacteria (S) (Figure 5B; Hodyra-Stefaniak et al. [26]). Moreover, within the model, the removal of the phage (P) by the innate immune system (I) would lead to a secondary increase in bacterial (S) count, indicating an inefficacy of phage therapy. This is in contrast to the available phage therapy related data [144,145,146,147,148]. Alternatively, this failure could be counteracted by adjusting the phage dose or changing the timing, as long as the interaction with the innate immunity is considered (Figure 5D). Nevertheless, this indicates a shortcoming of the current model described by Hodyra-Stefaniak, indicating further adapting of the model is needed to more closely reflect current knowledge of phage therapeutic outcomes.

## 6. Anti-Inflammatory Phage Properties Affect the Outcome of Phage Therapy

Most in-silico models miss one key feature—the interaction of the phage with the innate immune response. In theory, this interaction can be anti-inflammatory, leading to a suppression of the immune response, or pro-inflammatory, resulting in an increase of the immune response. Current literature states that phages are not able to induce pro-inflammatory responses [25]. Based on the recently described anti-inflammatory properties of phages [23,114], existing in-silico models can be further extended to include the interaction of phages with the innate immune response, as seen in the Appendix A.

By including the anti-inflammatory property of phages in the model, the prediction of the phage therapeutic outcome becomes successful again (Figure 5C). The phage (P) can, partly, subdue the innate immune response (I) and hence clears the bacterial infection (S). When a bacterial infection is combated with an initial high phage dose, the effects of the innate immune response are negligible (Figure 5D,E). Yet, if the phage has anti-inflammatory properties, the bacterial clearance occurs much faster according to the model. Nevertheless, when anti-phage antibodies (A) are present prior to the phage therapeutic intervention, the intervention fails as the phages are rapidly removed (Figure 5F,G).

When no pre-immunization to the phage is present, and no anti-inflammatory phage properties are considered, the removal of the bacterial infection is attributed mainly to the adaptive immune response against the bacteria (B) (Figure 5H). Initially, the phages (P) lead to a reduction of the bacterial count (S) but are themselves removed by a combination of the innate and adaptive immune response against the phage. This leads to a second rise in the bacterial concentration (S). In a later stage, the bacterial infection is removed by the adaptive immune response against the bacteria, hence the clearance of the bacterial infection is not due to the presence of the phage but due to the adaptive immune response against the bacteria (for the modeling purposes, the time of induction of specific antibodies was shorter than in physiological conditions). According to this model, when the anti-inflammatory properties of the phage (P) are considered, the bacterial infection (S) is cleared much faster and this is attributed to the presence of the phage (Figure 5I).

## 7. Relevance of Phage-Mammalian Host Immune Responses

The diverse ways in which phages interact with the human host are clear, and recently more work is being focused on this. Phage adherence to mucosal surfaces provides a previously unrecognized antimicrobial defense that actively protects the mucosal surface from bacterial infection [63,71]. This extension of the human immune system to include the action of symbiotic phages within the mucosal surface provides the eukaryotic host with a number of potential benefits. The phages offer a selective antimicrobial defense that operates at a much finer spectrum than some other broad-spectrum host secretions, such as the antibacterial lectin RegIII-ϒ [59]. Additionally, the interaction of phages with the mucosal layers can also lead to a higher rate of bacterial colonization in case of non-lytic filamentous phages [73,149]. The ability of phages to bind to mucus layers would provide them with a higher probability to contact and transcytose across epithelial cells [91]. This not only raises the question of whether they can interact with intracellular immune pathways but also whether phages could interact with mitochondria, which originated from a bacterial origin, once they are taken up by the cell. Although the presence of phages in mammalian cells has been observed [91,94], replication of these viruses in theses cell types has not yet been observed.

Phages can induce intra-cellular interactions with Toll-like receptors (TLR). TLR are responsible for the recognition of microbial and viral structures [43]. Viral nucleic acids act as pathogen associated molecular patterns (PAMPs) and are recognized by TLRs. It could thus be postulated that phage DNA might be recognized by TLR9, which is responsible for the recognition of viral DNA [150], after phagocytosis or transcytosis of the phage.

The observation that phages can directly interact with human immune cells and induce certain cytokine productions [23,114] has important implications for their use. Our in-silico model shows the positive effect of phage anti-inflammatory properties on the outcome of a bacterial infection, but these phage immune responses could have a much broader effect. Based on the anti-inflammatory responses observed by certain phages, it could even be postulated that phages could have an impact on allergic disorders such as asthma, rhinitis, and atopic dermatitis. The anti-inflammatory properties observed in certain phages could heighten their bacterial host’s fitness in an immunological context, creating potential microenvironments where the immune response is lowered [149], and the bacteria have a higher infection or survival rate. It is important to note that although the phage might have anti-inflammatory properties, this does not necessarily mean that the phage is able to effectively suppress the innate immune response. These anti-inflammatory properties do not seem comparable to typical immunosuppressive drugs or agents.

The most direct impact of phages might be during sepsis, where the lytic activity of the phage can reduce the bacterial burden and the immunomodulating properties of the phage could lead to a partial dampening of the inflammatory response induced by the bacteria or the bacterial lysis. Phage or phage-derived proteins that specifically interact with certain bacterial components (PAMPs) could even be used to moderate undesirable immune response (e.g., the use of phage T4 tail adhesin gp12 to capture and bind LPS in case of septic shock) [121]. The use of phages or phage-derived proteins as anti-inflammatory agents can lead to a possible new type of anti-inflammatory drugs with a new mode of action in comparison to the classic non-steroid anti-inflammatory drugs (NSAIDs). Possibly, these phages or phage-derived proteins might possess less side effects compared to NSAIDs. Phages can be engineered as nanocarriers for targeted drug delivery, or for the display of selected antigens and the subsequent stimulation of an immune response [24,151,152].

## 8. Conclusions and Areas of Future Investigation

The data reviewed here indicate that phages can interact with the mammalian immune system in a variety of ways that are both direct and indirect. However, the magnitude and nature of the influence that these viruses have on mammalian immunity are only beginning to come into focus. At present, the available data suggest that these interactions tend to be anti-inflammatory. If the observations by Van Belleghem et al. [23] and Zhang et al. [114] concerning the anti-inflammatory properties of phages can be further validated, it is conceivable that phages could influence both our interactions with our commensal flora as well as the outcome of phage therapeutic interventions.

However, the data on these interactions remains patchy, incomplete, and limited to small numbers of phages, cell types, and disease models. Further, definitive data indicating that phages impact human health or immunity, as opposed to cells or animal models, remains absent. Moreover, many of the specific mechanisms underlying the mammalian host immune response to phages remain unknown. Important areas of uncertainty include the following questions: How are phages taken up by cells? Is this an active or passive process? Is this uptake required to influence mammalian immunity or are cell surface interactions sufficient? Are these interactions specific to certain phages or phage families? Which parts of phages elicits the immune response? Do lytic and lysogenic phages influence host immunity in similar ways? Are these interactions primarily relevant to settings of immune interactions with commensal flora, microbial pathogens, or both? Knowing the answers to these and other questions could open many new fields of study and may facilitate the development of novel, phage-based therapies. We have much to learn but it is clear that phage and mammalian host interactions is an exciting and promising field of exploration.

## Figures and Tables

**Figure 1 viruses-11-00010-f001:**
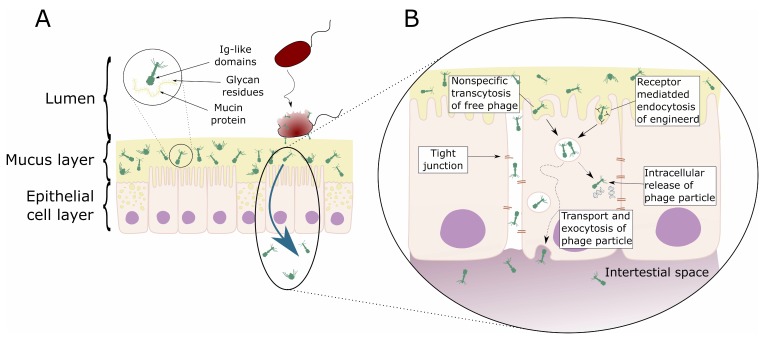
Schematic representation of direct interaction of phages with mammalian cells. (**A**) Bacteriophage adhering to mucus (BAM). Mucus is produced by the underlying epithelium. Phages of different morphologies (i.e., Myo-, Sipho-, and Podoviridae) can bind variable glycan residues displayed on mucin glycoproteins through variable capsid proteins, such as Ig-like domains. The adherence of phages to this mucus layer creates an antimicrobial layer that reduces bacterial attachment to and colonization of the mucus. This leads, in turn, to a reduction in epithelial cell death. Furthermore, these phages can migrate through theses epithelial cell layers subsequently ending up in the bloodstream. (**B**) Phage transcytosis. Binding interactions between phages and the membrane through transmembrane mucins, specific receptors, or through non-specific recognition, may allow signal transduction in the epithelial cell. Subsequently the phage particle is taken up by the epithelial cell. The internalized phage particles may be degraded leading to intracellular release of phage particles and DNA. Furthermore, it has been hypothesized that phage particles might cross the eukaryotic cell enabling phages to disseminate to the body. Phages may also gain access to the body via a “leaky gut”, where they bypass the epithelial cell barrier at sites of cellular damage or punctured vasculature. Figure adapted from Barr et al. [63,72].

**Figure 2 viruses-11-00010-f002:**
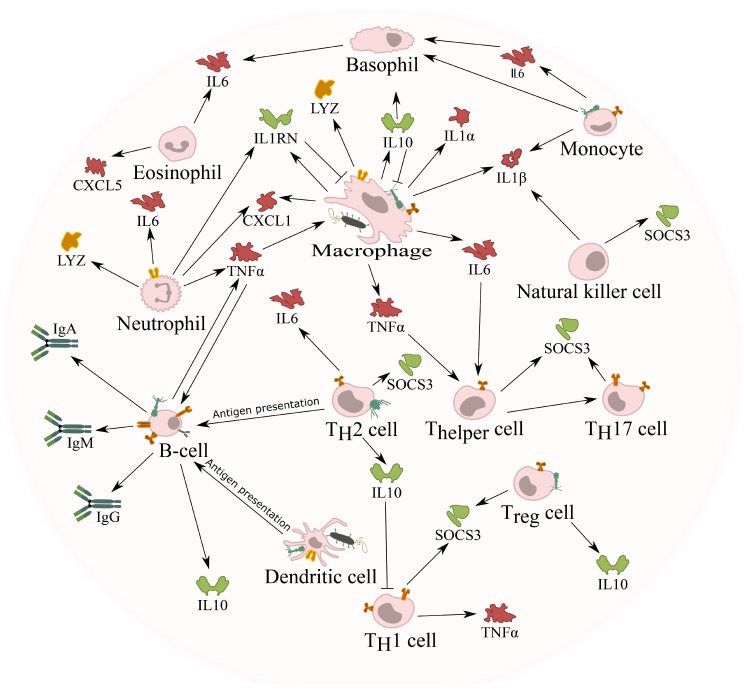
**Interaction of bacteriophages with mammalian immune cells.** Independent of the route of administration, phages can enter the bloodstream and tissues and encounter immune cells in the blood. Phages could encounter these immune cells whilst they are bound to their bacterial host and taken up together by either macrophages or dendritic cells. Alternatively, these phages can directly interact with any of these immune cells by either interacting with cell surface molecules or receptors, or taken up using a similar mechanism as observed with phage transcytosis. Once in contact with these immune cells, different pro- (red) or anti-inflammatory (green) cytokines are induced, giving the phage the opportunity to influence the immune response. For example, the induction of IL1RN by the phage blocks the pro-inflammatory signals induced by IL1α and IL1β. Although it is known that phages can induce cytokine response, the precise cells responsible are currently not known. Furthermore, the uptake of phages by antigen presenting cells (APC; e.g., dendritic cells) leads to the activation of B-cells and the production of specific antibodies against the phage.

**Figure 3 viruses-11-00010-f003:**
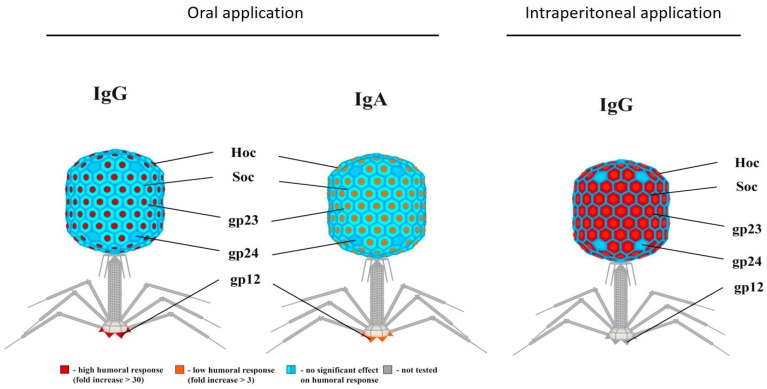
Antibody induction by phage T4 structural proteins. Individual contribution of T4 head proteins (Hoc, Soc, gp23, gp24, and gp12) to phage immunogenicity. Depending on the administration rote (i.e., oral or intraperitoneal), a difference in antibody response can be observed. When phages are administered orally, strong IgG or low IgA response towards Hoc can be observed, whereas intraperitoneal applications lead to high IgG responses towards Hoc and gp23. Modified Majewska et al. [24]. Permission was obtained for the reproduction of this figure.

**Figure 4 viruses-11-00010-f004:**
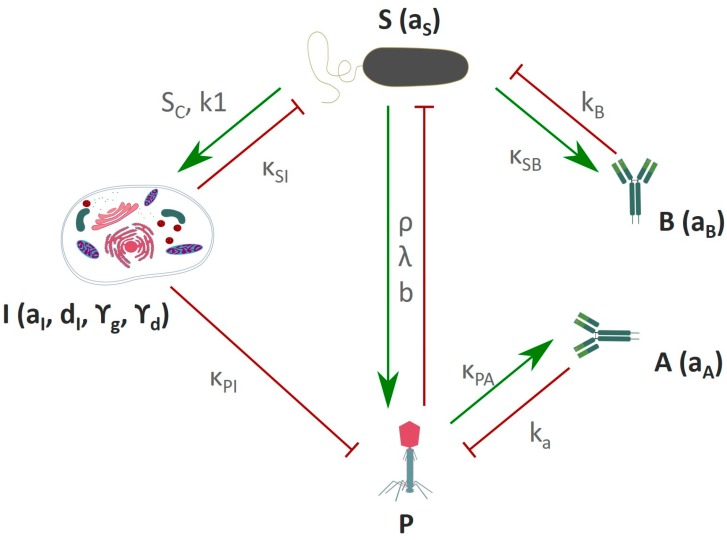
**Schematic representation of the immune response against phages and bacteria. P**–Phage, **S**–bacteria, **I**–innate immunity, **A**–adaptive immune response to phage, **B**–adaptive immune response to bacteria. Green arrows represent a stimulatory effect, red arrows represent an inhibitory effect. Variables and parameters used in these models are described in Appendix A. Adapted from Hodyra-Stefaniak et al. [26].

**Figure 5 viruses-11-00010-f005:**
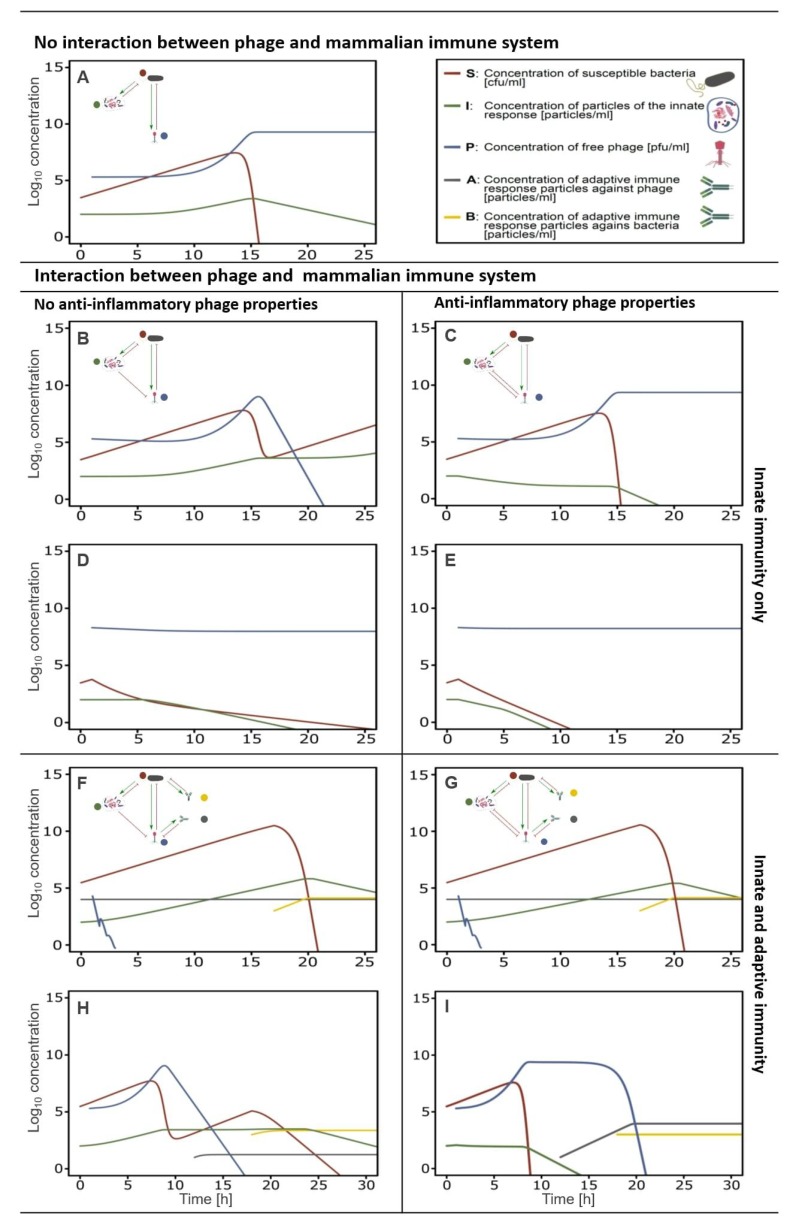
Effects of innate and adaptive immunity on the success or failure of phage antibacterial treatment, numerical simulations. Innate immune response. (**A**) No relation between innate immunity and phage viability. The survival of the phage is independent of the presence of an innate immune response. (**B**) Phage susceptibility to the innate immune response. The innate immunity has a negative effect on the phage survival and leads to its removal. Subsequently the bacteria are no longer infected by the phage, and a rise in bacteria is observed. (**C**) Phage susceptibility to the innate immune response, considering the anti-inflammatory property of the phage. The anti-inflammatory characteristic of the phage leads to a decline in innate immune particles. This has as effect that the bacterial count diminishes, and the phage survives, similar to A. (**D**) Phage susceptibility to innate immune response accommodated and counteracted by an increased phage dose. The higher phage dose leads to the removal of the pathogen and the survival of the phage. (**E**) Phage susceptibility to innate immune response accommodated and counteracted by an increased phage dose, considering the anti-inflammatory property of the phage. The effect is the same as in D, but the innate immune response is diminished. Innate and adaptive immune response. (**F**) Phage susceptibility to the innate immune response and presence of pre-immunization towards the phage. Presence of pre-existing anti-phage antibodies lead to a rapid drop in phage concentration, hence the phage has no effect on the survival of the bacteria. Once an adaptive immune response towards the bacteria is present, bacterial count decreases. (**G**) Phage susceptibility to the innate immune response and no pre-immunization to the phage exists, considering the anti-inflammatory property of the phage. The anti-inflammatory response of the phage has no direct influence on the phage survival in the presence of an adaptive immune response towards the phage. Overall the response is similar to F. (**H**) Phage susceptibility to the innate immune response and no pre-immunization to the phage exists. The absence of a specific adaptive immune response towards the phage leads to a decrease in the bacterial population. The combined effect of innate and adaptive immunity towards the phage leads to a drop-in phage particle concentration. (**I**) Phage susceptibility to the innate immune response and no pre-immunization to the phage exists, considering the anti-inflammatory property of the phage. Once the phage reaches a critical concentration (Pc, the concentration of phages needed to induce an anti-inflammatory response), the innate immune response decreases, and the phage concentration grows until all bacteria are removed. Once an adaptive immune response is present against the phage, the phage concentration diminishes until completely removed. Variables and parameters used in these models are described in Appendix A.

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
