# Peer review of "Interactions between Bacteriophage, Bacteria, and the Mammalian Immune System"

_viruses, 2018, doi:10.3390/v11010010_

Reviewer 1 Report

The review of Van Belleghem et al. refers to the interaction of bacteriophages and the mammalian immune system. The manuscript relies on an important aspect that causes concern and has been lately reviewed and referred by different studies focused on phage therapy.

Broad comments

The authors carry out a complete review of some topical factors associated with phage therapy. Based on a wide bibliography, they highlight some aspects as the interaction of bacteriophages with host cells, the phage transcytosis, and the different kinds of immune response (innate and adaptive) associated with phages or phage proteins. Which distinguish this review from other similar is perhaps the introduction of an in-silico model for explaining some of those issues. Although the discourse of this matter could be a little difficult to follow for a non-familiarized reader with this theme. The proposed model assumes that bacteriophages only produces an anti-inflammatory and not pro-inflammatory response in the host based on Miernikiewicz et al., work. That study is referred to T4 bacteriophage as a model. However, there is some discrepancy regarding this fact and some studies reference the induction of pro-inflammatory response based on cell experiments. In some cases, this response is attributed to a deficient purification of bacteriophage preparation. However, it is not enough investigated. Also, there is no agreement among in vitro and in vivo studies with respect to the induction or not of an immune response by the bacteriophages. Similarly, most studies refer to the immune response in a phage-bacteria environment. Nevertheless, bacteriophages can be also administered as a prophylactic measure and the single immune effect of phages are scarcely studied.

As authors recognize in their conclusions, there are still many aspects to be solved in this field.

This kind of review can help to prompt the researchers to delve into the study of the immunogenicity of bacteriophages.

 Minor comments

Please, check the nomenclature in italics of the microorganisms cited throughout the text and references. Also, for in vitro, and in vivo citations.

Line 190, it seems that the paragraph is not complete…” Cell perfusion and access, interaction with intracellular immune response…. Please, revise.

Please check and introduce the number of the reference after naming the author. If not, it is difficult to search the cite in the references list. It is the case of lines 239-240, 242-243, 276, 295, 305, and 437.

Author Response

We would like to thank the review for taking the time to go through the manuscript and provide us with helpful comments to improve our manuscript. 

The concerns raised by the reviewer are discussed below.

Please, check the nomenclature in italics of the microorganisms cited throughout the text and references. Also, for in vitro, and in vivo citations.

Thank you for bringing this to our attention. We have read through the manuscript and adapted this accordingly.

Line 190, it seems that the paragraph is not complete…” Cell perfusion and access, interaction with intracellular immune response…. Please, revise.

Thank you for pointing this out, this sentence should have been a subtitle. This must have been lost during formatting.

Please check and introduce the number of the reference after naming the author. If not, it is difficult to search the cite in the references list. It is the case of lines 239-240, 242-243, 276, 295, 305, and 437.

The reviewer makes a valid point, we did not include the number references to these parts of text as we think this is redundant information and would disturb the flow of the text.

Reviewer 2 Report

This review by Jonas D Van Belleghem and collaborators is totally timely and browses the various aspects of the interactions between bacteriophages and the mammalian immune system. Recent insights in this research field, including anti-inflammatory properties of phages and the induction of the mammalian immune response, are analysed, discussed and used to implement interaction modelling. The relevance of the various interactions of phages with mammals in term of bacteriophage therapeutic usage is also discussed. The work of many is described and well organized, making this review a great resource for those interested in phage therapy.

 I only have a few comments and suggestions that could be addressed before publication.

 l. 15: “immense” numbers is a bit vague.

l. 16: bacteriophages in general are not only parasites as they (temperate) can also provide new properties to the host. “infect” is probably more appropriate as it does not bet on the infection outcome.

l. 27: colonize and live in symbiosis

l. 31: have been a primary focus (other co-exist!)

l. 39, 40: The list numbering should be changed as it looks like reference numbering.

l. 43: underlying the important…

l. 47: the gut microbiome/virome literature is far more abundant than cited, I obviously understand that not all articles can be cited, however I’d like to have a rationale for choosing the ones that are cited. In addition, this recent review could be also cited : Muniesa M et al, Front Microbiol 2017.

l. 50: Not all temperate integrate their genome, some remain as episomes.

l. 59: 109 or 10^9 ?

l. 92: mucin glycoproteins that are secreted

l. 78: the first sentence of the paragraph could be a bit more detailed: what kind of immunity is stimulated ?

l. 112-118: Ig-like domains. It is stated that the phages using Ig-like domains are probably under positive selection, this should increased their proportion above 25% ?

l. 115: Phages that utilize

l. 150: what’s the difference between constant and continual ?

l. 158: rephrasing could be done

l. 190: Phrase fragment or sub-title ?

l. 158: rephrasing could be done

l. 206: is the term perfusion appropriate here ?

l. 207-208: If phage particles are degraded, how can they be visualized and interact with the eukaryotic cell?

l. 226: Is there a reference for the accumulation in spleen and liver ?

l. 230: Glu-Lys substitution (not transition); please explain the double charged change if only one residue is changed

l. 232: Should re-cite the organs before writing “both organs”

l. 245: induced by phages

l. 268: please explain opsonisation and its effects

l. 278: but the precise

l. 289: Fig. 3 should be cited

l. 302: I haven’t found the meaning of the asterisks, Fig. 3 should be cited here as well

l. 325: please rephrase “cannot be considered as comparable to physiology”

l. 335: The mode of administration should be indicated

l. 358-359 : the sentence starts with « most studies… » but only one is cited (ref 128) ?

l. 362: this is the first time in the review that an adjuvant is mentioned, are they any of the cited studies that use adjuvant? If yes, please mention and precise the nature of the adjuvant.

Figure 4: The legend should describe the constants and variables or refer to Table S2. I would indicate “Innate” and “Adaptive” on the figure.

Author Response

We would like to thank the review for taking the time to go through the manuscript and provide us with helpful comments to improve our manuscript. 

The concerns raised by the reviewer are discussed below.

l. 15: “immense” numbers is a bit vague.

We replaced the word with large, in the hope this clarifies the statement more.

l. 16: bacteriophages in general are not only parasites as they (temperate) can also provide new properties to the host. “infect” is probably more appropriate as it does not bet on the infection outcome.

We changed this as recommended by the reviewer.

l. 27: colonize and live in symbiosis

We added the suggestion to the main text.

l. 31: have been a primary focus (other co-exist!)

As suggested by the reviewer we altered the text.

l. 39, 40: The list numbering should be changed as it looks like reference numbering.

Thank you for bringing this to our attention, we replaced the “(“ with “[“.

l. 43: underlying the important…

The sentence was altered as suggested by the reviewer.

l. 47: the gut microbiome/virome literature is far more abundant than cited, I obviously understand that not all articles can be cited, however I’d like to have a rationale for choosing the ones that are cited. In addition, this recent review could be also cited : Muniesa M et al, Front Microbiol 2017.

The section the reviewer refers to describes the specific microenvironments where phages have been detected. For each microenvironment that has been studied the corresponding literature was cited. For this section we referred to the specific scientific research and data. The reference mentioned by the reviewer, although very interesting does in our opinion not fit in this section. The aforementioned review gives general information of phages in the body but does not discuss specific microenvironments.

l. 50: Not all temperate integrate their genome, some remain as episomes.

We added the recommended information.

l. 59: 109 or 10^9 ?

Thank you for bringing this to our attention.

l. 92: mucin glycoproteins that are secreted

Thank you for bringing this grammatical error to our attention. This has been corrected.

l. 78: the first sentence of the paragraph could be a bit more detailed: what kind of immunity is stimulated ?

As the induced immune response could both be pro- or anti-inflammatory we decided to keep this general statement.

l. 112-118: Ig-like domains. It is stated that the phages using Ig-like domains are probably under positive selection, this should increased their proportion above 25% ?

Of all known phages 25% seem to posses Ig-like domains in their capsids. The presence of these structures could lead to a positive selection of these phages. To our knowledge such an enrichment has not been reported to data.

l. 115: Phages that utilize

Thank you for pointing out this grammatical error. This has been corrected.

l. 150: what’s the difference between constant and continual ?

By constant we meant that once it is in contact it stays in contact, whereas continual we meant that these interactions happen over different times. We understand that this is not really clear, therefor we omitted ‘continual’ from this sentence.

l. 158: rephrasing could be done

The sentence was rephrased as “In vitro uptake of phage M13 could be blocked by chloroquine, an inhibitor known to block clathrin-dependent endocytosis, suggesting this was the proposed pathway for internalization”

l. 190: Phrase fragment or sub-title ?

Due to formatting this subtitle lost its layout. We have corrected this accordingly.

l. 206: is the term perfusion appropriate here ?

Perfusion is the passage of fluid through the circulatory system or lymphatic system to an organ or a tissue, usually referring to the delivery of blood to a capillary bed in tissue. In that sense perfusion might not be the perfect wording. Here perfusion means that the cell is completely ‘soaked’ with phage, in the sense that phage are massively taken up by the eukaryotic cell.

l. 207-208: If phage particles are degraded, how can they be visualized and interact with the eukaryotic cell?

The degradation of the phage particle could be time dependent, depending on the timepoints used to visualize the phage the phage could either still be intact or degraded. Additionally, many of these experiments make use of fluorescently labeled phage. In that case, if the phage were degraded the fluorescent label might still be intact to show the location of the phage.

l. 226: Is there a reference for the accumulation in spleen and liver ?

there is indeed a very old reference about the accumulation of phages in the liver and spleen from 1934: “W.J. Nungester & R.M. Watrous. 1934. Accumulation of Bacteriophage in Spleen and Liver Following Its Intravenous Inoculation. Experimental Biology and Medicine”.

We thank the reviewer to point this out, and we included the reference in the manuscript.

l. 230: Glu-Lys substitution (not transition); please explain the double charged change if only one residue is changed

The reviewer is correct to point out that there is indeed only one charge change instead of two. We adapted this accordingly in the MS.

l. 232: Should re-cite the organs before writing “both organs”

We altered this in the manuscript.

l. 245: induced by phages

We thank the reviewer to point out this symantic difference and changed this in the manuscript.

l. 268: please explain opsonisation and its effects

We have clarified this more in the text.

l. 278: but the precise

As suggested by the reviewer this was adapted in the text.

l. 289: Fig. 3 should be cited

Figure 3 describes the different phage epitopes that could elicit an antibody response. Although the whole phage particle is represented, we understand the confusion of the reviewer that it would be of interest to link the observation of gp12 to bind LPS to this figure. As this is out of the scoop of this figure we decided to not reference this figure in this sentence.

l. 302: I haven’t found the meaning of the asterisks, Fig. 3 should be cited here as well

The asterisks in gp23* and gp24* have no specific reference to something additional but are, as far as we know, part of the proteins name.

Also this sentence refers to the ability of these proteins to induce cytokine responses, whereas Figure 3 refers to antibody responses towards these proteins.

l. 325: please rephrase “cannot be considered as comparable to physiology”

We altered this sentence in the hope it clarifies the statement more.

l. 335: The mode of administration should be indicated

The mode of administration was intravenous injection, we added this information to the text.

l. 358-359 : the sentence starts with « most studies… » but only one is cited (ref 128) ?

We added the correct references.

l. 362: this is the first time in the review that an adjuvant is mentioned, are they any of the cited studies that use adjuvant? If yes, please mention and precise the nature of the adjuvant.

It is well known to be able to induce a potent antibody response to a specific epitope during vaccination adjuvant need to be added. We did not elaborate on this fact because we think this is outside the scoop of this review. To our knowledge there have currently been no studies of phage induced antibody responses that include adjuvant, the reason for this is that for phage therapeutic purposes most researchers are interested in whether the phage on its own can induce an antibody response instead of deliberately inducing anti-phage antibodies for vaccination purposes.

Figure 4: The legend should describe the constants and variables or refer to Table S2. I would indicate “Innate” and “Adaptive” on the figure.

The reviewer is correct to point this out, we added this information to the figure legend.